# Closer to critical resting-state neural dynamics in individuals with higher fluid intelligence

Takahiro Ezaki[1,2], Elohim Fonseca dos Reis [3], Takamitsu Watanabe[4,5], Michiko Sakaki [6,7] & Naoki Masuda [3,8,9]*

According to the critical brain hypothesis, the brain is considered to operate near criticality and realize efficient neural computations. Despite the prior theoretical and empirical evidence in favor of the hypothesis, no direct link has been provided between human cognitive performance and the neural criticality. Here we provide such a key link by analyzing resting-state dynamics of functional magnetic resonance imaging (fMRI) networks at a whole-brain level. We develop a data-driven analysis method, inspired from statistical physics theory of spin systems, to map out the whole-brain neural dynamics onto a phase diagram. Using this tool, we show evidence that neural dynamics of human participants with higher fluid intelligence quotient scores are closer to a critical state, i.e., the boundary between the paramagnetic phase and the spin-glass (SG) phase. The present results are consistent with the notion of "edge-of-chaos" neural computation.

[1] PRESTO, Japan Science and Technology Agency, Kawaguchi, Saitama, Japan. [2] Research Center for Advanced Science and Technology, The University of Tokyo, Meguro-ku, Tokyo, Japan. [3] Department of Engineering Mathematics, University of Bristol, Clifton, Bristol, UK. [4] Institute of Cognitive Neuroscience, University College London, 17 Queen Square, London WC1N 3AZ, UK. [5] RIKEN Center for Brain Science, Wako, Saitama, Japan. [6] School of Psychology and Clinical Language Sciences, University of Reading, Earley Gate, Whiteknights Road, Reading, UK. [7] Research Institute, Kochi University of Technology, Kami, Kochi, Japan. [8] Department of Mathematics, University at Buffalo, State University of New York, Buffalo, New York, USA. [9] Computational and Data-Enabled Science and Engineering Program, University at Buffalo, State University of New York, Buffalo, New York, USA. *email: naokimas@buffalo.edu

Critical brain hypothesis posits that the brain operates near a critical regime, i.e., boundary between different phases showing qualitatively different behaviors[1–6]. This hypothesis has been investigated for more than two decades including criticisms such as the presence of alternative mechanisms explaining power law scaling in the relevant observables[7–10]. Experimental evidence such as the recovery of critical behavior after interventions, which is difficult to explain by alternative mechanisms, lends supports to the hypothesis[9].

Theoretical and experimental work has shown that neural systems operating near criticality are advantageous in information transmission, information storage, classification, and nonlinear input filtering[1,3,5,11–14]. These findings align with the idea of edge-of-chaos computation, with which computational ability of a system is maximized at a phase transition between a chaotic phase and a nonchaotic phase[15–17]. These findings are also in line with a general contention that cognitive computations occur as neural dynamical processes[18,19].

A prediction from the critical brain hypothesis is that neural dynamics in individuals with higher cognitive abilities should be closer to criticality than in those with lower cognitive abilities. However, whether high cognitive skills are associated with criticality has not been empirically proven. In fact, recent emerging evidence suggests that human cognitive performance is associated with appropriate transitions between relatively discrete brain states during rest[20–22], working memory tasks[23], and visual perception tasks[24]. Furthermore, these and other studies[18,19,25] support that state-transition dynamics in the brain involve large-scale brain networks. These arguments are consistent with the proposal that many cognitive functions seem to depend on network connectivity among various regions scattered over the whole brain[26]. On these grounds, in the present study we hypothesize that complex and transitory neural dynamics of the brain network (i.e., dynamic transitions among discrete brain states) that are close to criticality are associated with high cognitive performance of humans.

Two major conventional methods for examining criticality and edge-of-chaos computation in empirical neural data are not capable of testing this hypothesis for their own reasons. First, many of the experimental studies testing the critical brain hypothesis have examined neuronal avalanches[11,12], including the case of humans[5,27,28]. Neuronal avalanches are bursts of cascading activity of neurons, whose power-law properties have been related to criticality. However, studies of neuronal avalanches have focused on their scale-free dynamics in space and time, with which statistics of avalanches obey power laws. Scale-free dynamics of neuronal avalanches is a question orthogonal to patterns of transitions between discrete states. Second, nonlinear time series analysis has found that electroencephalography signals recorded from the brains of healthy controls are chaotic and that the degree of chaoticity is stronger for healthy controls than individuals with, for example, epilepsy, Alzheimer's disease, and schizophrenia[29]. However, this method is not usually for interacting time series. Therefore, it does not directly reveal how different brain regions interact or whether possible critical or chaotic dynamics are an outcome of the dynamics at a single region or interaction among different regions.

In the present study, we develop a data-driven method to measure the extent to which neural dynamics obtained from large-scale brain networks are close to criticality and complex state-transition dynamics. The method stands on two established findings. First, statistical mechanical theory of the Ising spin-system model posits that the so-called spin-glass phase corresponds to rugged energy landscapes (and therefore, complex state-transition dynamics)[30] and chaotic dynamics[31–33]. Therefore, we are interested in how close the given data are to dynamics in the spin-glass phase. Second, the Ising model has been fitted to various electrophysiological data[6,34–36] and fMRI data recorded from a collection of regions of interest (ROIs)[20,21,24,37,38] during rest or tasks with a high accuracy. Therefore, we start by fitting the Ising model to the multivariate fMRI data. Then, we draw phase diagrams of functional brain networks at a whole-brain level. By construction, the dynamical behavior of the system is qualitatively distinct in different phases. The method determines the location of a brain in the phase diagram and thus tells us whether the large-scale brain dynamics of individual participants are ordered, disordered, or chaotic (i.e., spin-glass) dynamics as well as how close the dynamics are to a phase transition curve, on which the system shows critical behavior.

We deploy this method to resting-state fMRI data recorded from human adults with different intelligence quotient (IQ) scores. As a cognitive ability of interest, we focus on fluid intelligence, which refers to the ability to think logically and solve problems with a limited amount of task-related information[39]. Fluid intelligence is strongly related to the general intelligence factor, $g$[39] and predictive of real-world outcomes such as job performance[40]. We examine our hypothesis that large-scale brain dynamics of individuals higher in the intelligence score that measures fluid intelligence are closer to critical.

## Results
**Brain dynamics are close to the spin-glass phase transition.** We first fitted the pairwise maximum entropy model (PMEM), which assumes pairwise interaction between ROIs and otherwise produces a maximally random distribution, which is a Boltzmann distribution. The PMEM is equivalent to the inverse Ising model, where the parameters of the Ising model are inferred from data. Because the model assumes binary data, we binarized the resting-state fMRI signals obtained from 138 healthy adults. The binarized activity pattern at $N(= 264)$ ROIs[41] at time $t$ ($t = 1, \dots, t_{\max}$; $t_{\max} = 258$) is denoted by $\mathbf{S}(t) = (S_1(t), \dots, S_N(t)) \in \{-1, +1\}^N$, where $S_i(t) = 1$ and $S_i(t) = -1$ ($i = 1, \dots, N$) indicate that ROI $i$ is active (i.e., the fMRI signal is larger than a threshold) and inactive (i.e., smaller than the threshold), respectively. We fitted the following probability distribution to the population of the 138 participants by maximizing a pseudo likelihood (see "Methods")[24,34]:

$$P(\mathbf{S}|\mathbf{h}, \mathbf{J}) = \frac{\exp[-E(\mathbf{S}|\mathbf{h}, \mathbf{J})]}{\sum_{\mathbf{S} \in [-1,1]^N} \exp[-E(\mathbf{S}|\mathbf{h}, \mathbf{J})]}. \tag{1}$$

In Eq. (1),

$$E(\mathbf{S}|\mathbf{h}, \mathbf{J}) = -\sum_{i=1}^{N} h_i S_i - \frac{1}{2} \sum_{i=1}^{N} \sum_{j=1, j \neq i}^{N} J_{ij} S_i S_j \tag{2}$$

is the energy of activity pattern $\mathbf{S}$, $\mathbf{h} = \{h_i : 1 \leq i \leq N\}$, and $\mathbf{J} = \left\{ J_{ij} : 1 \leq i \neq j \leq N \right\}$, where $J_i = J_{ji}$. Although we refer to $E$ as the energy, $E$ does not represent the physical energy of a neural system but is a mathematical construct representing the frequency with which activity pattern $\mathbf{S}$ appears in the given data. Activity pattern $\mathbf{S}$ appears rarely in the data if $E$ corresponding to $\mathbf{S}$ is large and vice versa. Parameter $h_i$ represents the tendency that $S_i = 1$ is taken because a positive large value of $h_i$ implies that $S_i = 1$ as opposed to $S_i = -1$ lowers the energy and hence raises the probability that $\mathbf{S}$ with $S_i = 1$ appears. Parameter $J_{ij}$ represents a functional connectivity between ROIs $i$ and $j$ because, if $J_{ij}$ is away from 0, $S_i$, and $S_j$ would be correlated in general. We denote the estimated parameter values by $\hat{\mathbf{h}}$ and $\hat{\mathbf{J}}$.

Then, to evaluate how close the current data are to criticality, we drew phase diagrams by sweeping values of $\mathbf{J}$. In the phase diagrams, we fixed $\mathbf{h}$ at $\hat{\mathbf{h}}$ following the theoretical convention[30],

including when the PMEM is applied to data analysis[6]. We set $\mathbf{h} = \hat{\mathbf{h}}$ also because changing the $\mathbf{h}$ values did not qualitatively change the phase diagrams (Supplementary Fig. 1). Then, we varied the mean $\mu$ and standard deviation $\sigma$ of $\mathbf{J}$ by linearly transforming $\mathbf{J}$, i.e.,

$$J_{ij} = (\hat{J}_{ij} - \hat{\mu})\frac{\sigma}{\hat{\sigma}} + \mu. \qquad (3)$$

In (3), $\hat{\mu} = 1.57 \times 10^{-3}$ and $\hat{\sigma} = 3.57 \times 10^{-2}$ are the mean and standard deviation of the off-diagonal elements of $\hat{\mathbf{J}}$ estimated for the empirical data. We chose the parametrization given in Eq. (3) motivated by the past investigation of the archetypical Sherrington-Kirkpatrick (SK) model of spin systems[30]. The SK model, a type of Ising model, is defined with parameters $J_{ij}$ ($1 \le i \ne j \le N$) that are independently drawn from the Gaussian distribution with the tunable mean and standard deviation and has extensively been studied for investigating the so-called spin-glass phase transition owing to its theoretical tractability. In the spin-glass phase, the system shows a disorderly frozen pattern of spins rather than uniform or periodic ones. For each set of $J_{ij}$ values ($1 \le i \ne j \le N$) specified by a ($\mu$, $\sigma$) pair, we performed Monte Carlo simulations and calculated observables (see "Methods"). In this manner, we generated a phase diagram for each observable in terms of $\mu$ and $\sigma$.

Two primary observables (called order parameters in physics literature) employed in studies of spin systems are the magnetization, denoted by $m$, and the spin-glass order parameter, denoted by $q$. The magnetization is defined by $m = \sum_{1 \le i \le N}\langle S_i \rangle / N$, where $\langle \cdot \rangle$ represents the ensemble average, and quantifies the mean tendency that $S_i = 1$ as opposed to $S_i = -1$ is taken across the ROIs. The spin-glass order parameter is defined by $q = \sum_{1 \le i \le N}\langle S_i \rangle^2 / N$ and represents the degree of local magnetization at individual ROIs. We show $m$ and $q$ as functions of $\mu$ and $\sigma$ in Fig. 1a, b, respectively. The obtained phase diagrams were qualitatively the same as those for the SK model of the same system size, which was given by Eqs. (1) and (2) with each $J_{ij}$ ($=J_{ji}$, $i \ne j$) being independently drawn from a Gaussian distribution with mean $\mu$ and standard deviation $\sigma$ (Fig. 1e, f). The parameter space is composed of three qualitatively different phases[30]. The paramagnetic phase, characterized by $m = 0$ and $q = 0$ in the limit of $N \to \infty$, represents the situation in which each $S_i$ randomly flips between 1 and $-1$, yielding no magnetization. The ferromagnetic phase, characterized by $m \ne 0$ and $q > 0$, represents the situation in which (almost) all $S_i$'s align in one direction (i.e., $S_i = 1$ or $S_i = -1$). The spin-glass (SG) phase, characterized by $m = 0$ and $q > 0$, represents the situation in which each $S_i$ is locally magnetized but not globally aligned to a specific direction[30]. Note that the finite size effect of our system blurred the boundaries between the different phases. The current data pooled across the participants lie in the paramagnetic phase and are close to the boundary to the SG phase (crosses in Fig. 1a, b). In theory, the spin-glass susceptibility, $\chi_{SG} = N^{-1}\beta^2 \sum_{1 \le i,j \le N} c_{ij}^2$, where $c_{ij} = \langle S_i S_j \rangle - m_i m_j$, diverges on the boundary between the paramagnetic and SG phases[30]. The empirical data yielded a relatively large $\chi_{SG}$ value in the phase diagram (Fig. 1c). In contrast, we did not find a signature of phase transition in terms of the uniform suscept-ibility defined by $\chi_{uni} = N^{-1}\beta \sum_{1 \le i,j \le N} c_{ij}$, which characterizes the transition between the paramagnetic and ferromagnetic phases[30] (Fig. 1d). Note that the phase diagrams for $\chi_{SG}$ and $\chi_{uni}$ resemble those obtained from the SK model (Fig. 1g, h).

Next, we examined where brain activity patterns of each participant were located in the phase diagrams. We did so by finding the $\mu$ and $\sigma$ values corresponding to the $\chi_{SG}$ and $\chi_{uni}$ values of each participant (see "Methods"). It should be noted

that $\chi_{SG}$ and $\chi_{uni}$ can be calculated for each individual only from the covariance matrix of the data, without estimating the PMEM. The location of each participant in the phase diagram of $\chi_{SG}$ is shown by the circles in Fig. 1c. The cross section of this phase diagram for $\mu = \hat{\mu}$ (along the dashed line shown in Fig. 1c) is shown in Fig. 1i. We also projected the $\chi_{SG}$ values for the individual participants (circles in Fig. 1i) based on the value of $\sigma$ estimated for each individual (circles in Fig. 1c). Figure 1i suggests that the empirical data are located in a range of $\sigma$ that constitutes a peak, further confirming that the brain dynamics of different participants are close to the paramagnetic–SG phase transition and to different extents. In contrast, the participants were far from the paramagnetic–ferromagnetic phase boundary. This is confirmed in Fig. 1j, which is a cross section of the phase diagram for $\chi_{uni}$ (along the dashed line shown in Fig. 1d) together with the $\chi_{uni}$ values for the single participants.

The $\chi_{SG}$ value for the individual participants was off the largest possible values in the phase diagram (Fig. 1i). To examine this point, we carried out a finite size scaling on $\chi_{SG}$ (Fig. 1k). To emulate systems of smaller sizes than $N = 264$, we selected $N'$ out of the $N$ ROIs uniformly at random and fitted the PMEM. The estimated parameter values are denoted by $\hat{\mathbf{h}}$ and $\hat{\mathbf{J}}$ without confusion. Then, we simulated the equilibrium state of the system by scanning $\mathbf{J}$ according to Eq. (3), where we varied $\sigma$ while fixing $\mu = \hat{\mu}$. In this manner, we sought to investigate how close the data were to the SG phase transition at each $N'$ value. As shown in Fig. 1k, the peak value of $\chi_{SG}$ increased as $N'$ increased, suggesting that the paramagnetic–SG phase transition is approached as the system size increases. In addition, the position of the peak, denoted by $\sigma_{peak}$, shifted toward the value for the empirical data, $\hat{\sigma}$, as $N$ increased. By regressing $\sigma_{peak}/\hat{\sigma}$ linearly on $1/N'$ (inset of Fig. 1k), we estimated $\sigma_{peak}/\hat{\sigma} = 1.45 \pm 0.04$ in the limit $N' \to \infty$.

**The performance IQ is associated with the criticality**. To test our hypothesis that criticality of brain dynamics is associated with human fluid intelligence, we examined the correlation between $\chi_{SG}$, which encodes the proximity of each participant's neural dynamics to the paramagnetic–SG phase transition (Fig. 1c, i), and the performance IQ score. The performance IQ score is defined based on tasks that are reflective of fluid intelligence[42,43]. An enlargement of Fig. 1c is shown in Fig. 2a, where the participants are shown in different colors depending on whether they have a higher performance IQ score (defined by the score value larger than or equal to the median, 109, $n = 68$) and a lower score ($n = 63$). We found that higher-IQ participants tended to be closer to the paramagnetic–SG phase transition than lower-IQ participants, as measured by $\sigma$ ($t_{129} = 3.17$, $P < 0.002$, Cohen's $d = 0.55$ in a two-sample $t$ test). The results were qualitatively the same when the outliers were excluded ($t_{127} = 3.52$, $P < 10^{-3}$, $d = 0.62$). In contrast, the two groups were not different in terms of the distance to the paramagnetic–ferromagnetic phase transition as measured by $\mu$ ($t_{129} = 0.77$, $P = 0.44$, $d = 0.13$ with the outlier included; $t_{127} = 0.85$, $P = 0.40$, $d = 0.15$ with the outlier excluded).

More systematically, we found a mild positive correlation between $\chi_{SG}$ and the performance IQ score ($r_{129} = 0.24$, $P_{Bonferroni} = 0.011$; also see Fig. 2b). However, the verbal IQ score, which is based on individuals' verbal knowledge[42,43], was not correlated with $\chi_{SG}$ ($r_{126} = 0.06$, $P_{uncorrected} = 0.50$, Fig. 2c). The correlation between $\chi_{SG}$ and the performance IQ score was also significantly larger than the correlation between $\chi_{SG}$ and the verbal IQ score ($t_{121} = 2.33$, $P = 0.021$, in the Williams $t$ test for comparing two nonindependent correlations with a variable in common[44]). These results suggest that the criticality of brain dynamics plays more roles in fluid intelligence than when simply retrieving verbal knowledge. Note that we partialed out the effects

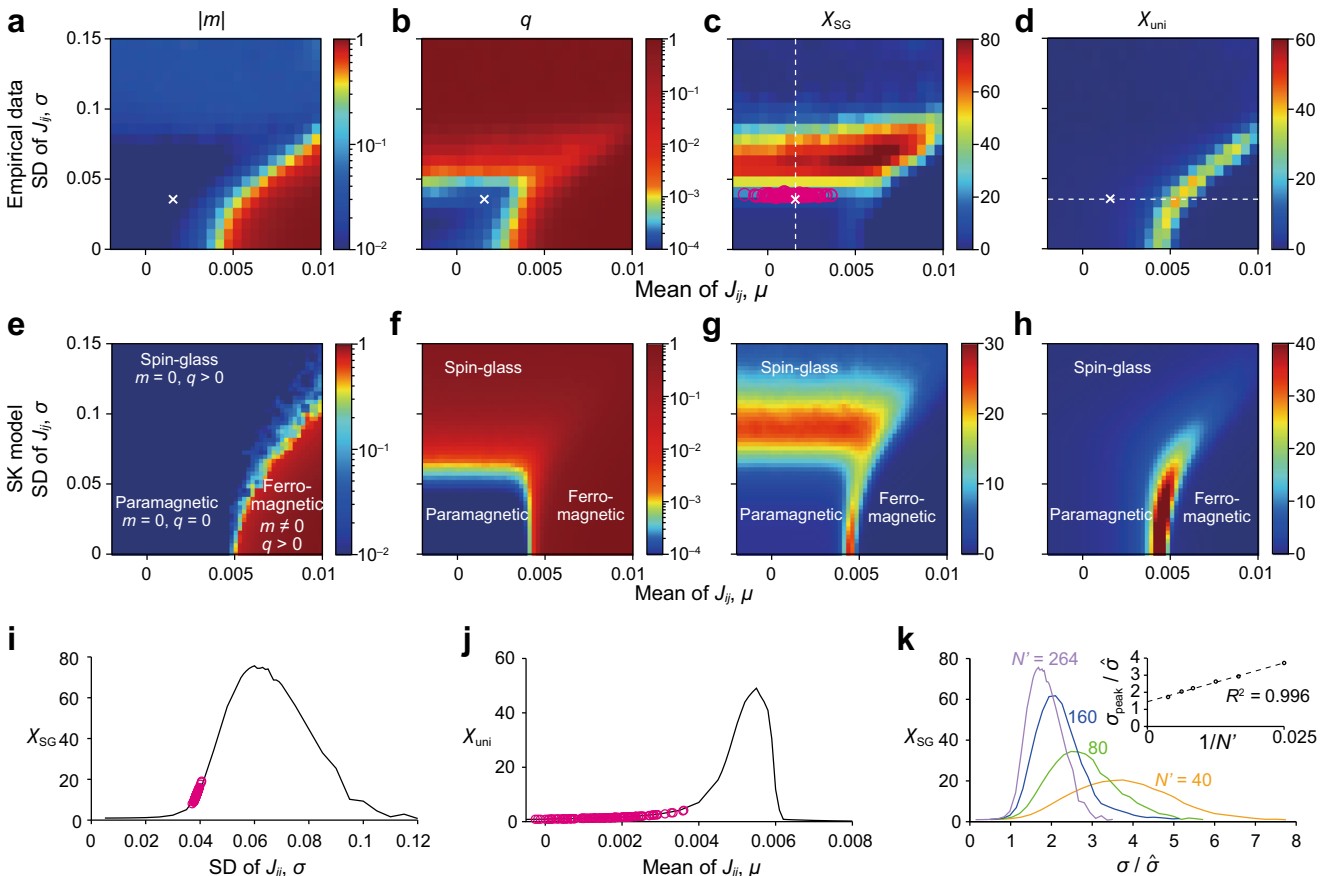

**Fig. 1 a–d** Phase diagrams for the empirical data. **e–h** Phase diagrams for the SK model. **a, e**: $|m|$. **b, f**: $q$. **c, g**: $\chi_{SG}$. **d, h**: $\chi_{uni}$. In **a, d** the crosses represent the mean and standard deviation of the $J_{ij}$ estimated for the entire population of the participants, i.e., $(\hat{\mu}, \hat{\sigma})$. In **c** a circle represents a participant. In **a** and **e** we plot $|m|$ instead of $m$. This is because averaging over simulations and over realizations of **J** would lead to $m \approx 0$ due to symmetry breaking, even if $m \neq 0$ in theory such as in the ferromagnetic phase. **i** $\chi_{SG}$ as a function of $\sigma$, with $\mu = \hat{\mu}$ being fixed. **j** $\chi_{uni}$ as a function of $\mu$, with $\sigma = \hat{\sigma}$ being fixed. In **i** and **j**, the curves are the cross-sectional view of **c** and **d**, respectively, along the dashed line in **c** or **d**. The circles in **i** and **j** represent the individual participants and are the projection of the circles in **c** and **d** onto the dashed line. **k** Scaling behavior of $\chi_{SG}$ when the system size $N'$ is varied. The value of $\sigma = \sigma_{peak}$ that maximizes $\chi_{SG}$ is plotted against $1/N'$ in the inset. The dashed line is the linear regression based on the six data points, $N' = 40, 60, 80, 120, 160,$ and $264$. The coefficient of determination is denoted by $R^2$.

of the age and gender in this and the following analysis unless we state otherwise.

The correlation between the full IQ score[42,43] and $\chi_{SG}$ was intermediate between the results for the performance and verbal IQ scores ($r_{130} = 0.19$, $P = 0.026$; also see Fig. 2d), which is natural because the performance and verbal IQ scores are components of the full IQ score.

The association between the spin-glass susceptibility, $\chi_{SG}$, and the different types of IQ scores were robust in the following four ways. First, the exclusion of the two outliers determined by Tukey's 1.5 criteria[45] did not affect the significance of the results ($\chi_{SG}$ vs performance IQ: $r_{127} = 0.27$, $P_{Bonferroni} = 0.005$; $\chi_{SG}$ vs verbal IQ: $r_{124} = 0.13$, $P_{Bonferroni} = 0.27$; $\chi_{SG}$ vs full IQ: $r_{128} = 0.25$, $P = 0.005$). Second, the results were robust against variation on the threshold value for binarizing the fMRI signal (Supplementary Fig. 2). Furthermore, changes in the threshold value did not substantially alter the phase diagrams (Supplementary Fig. 3). Third, the results were preserved even when the global signal (see Methods) was not subtracted from the fMRI signals ($\chi_{SG}$ vs performance IQ: $r_{129} = 0.22$, $P_{Bonferroni} = 0.02$; $\chi_{SG}$ vs verbal IQ: $r_{126} = 0.046$, -$P_{uncorrected} = 0.61$; $\chi_{SG}$ vs full IQ: $r_{130} = 0.18$, $P = 0.043$; the outliers were not removed). Fourth, we did not find a gender difference in the correlation coefficient between $\chi_{SG}$ and the IQ scores

(performance IQ: $Z = 0.33$, $P = 0.74$ in a $Z$-test for a pair of correlation coefficients[46]; verbal IQ: $Z = 0.43$, $P = 0.67$; full IQ: $Z = 0.17$, $P = 0.86$). In this gender-difference analysis, we partialed out the effect of the age but not the gender.

**Irrelevance of the paramagnetic–ferromagnetic transition.** The IQ was not correlated with $\chi_{uni}$ (performance IQ: $r_{129} = 0.10$, $P_{uncorrected} = 0.27$; verbal IQ: $r_{126} = 0.093$, $P_{uncorrected} = 0.30$; full IQ: $r_{130} = 0.10$, $P = 0.24$, each test including the outliers; performance IQ: $r_{124} = 0.013$, $P_{uncorrected} = 0.89$; verbal IQ: $r_{121} = 0.039$, $P_{uncorrected} = 0.67$; full IQ: $r_{125} = 0.020$, $P = 0.82$, each test excluding the outliers). The specific heat (denoted by $C$; see "Methods" for definition) was only mildly correlated with the performance IQ score (performance IQ: $r_{129} = 0.21$, $P_{Bonferroni} = 0.034$; verbal IQ: $r_{126} = -0.0056$, $P_{uncorrected} = 0.95$; full IQ: $r_{130} = 0.13$, $P = 0.14$, each test including the outliers; performance IQ: $r_{125} = 0.16$, $P_{uncorrected} = 0.08$; verbal IQ: $r_{122} = -0.016$, $P_{uncorrected} = 0.86$; full IQ: $r_{126} = 0.10$, $P = 0.26$, each test excluding the outliers). Because $\chi_{uni}$ and $C$ diverge in the paramagnetic–ferromagnetic phase transition but not in the paramagnetic–SG phase transition[30], these negative results lend another support to the relevance of the SG phase rather than the ferromagnetic phase to intelligence.

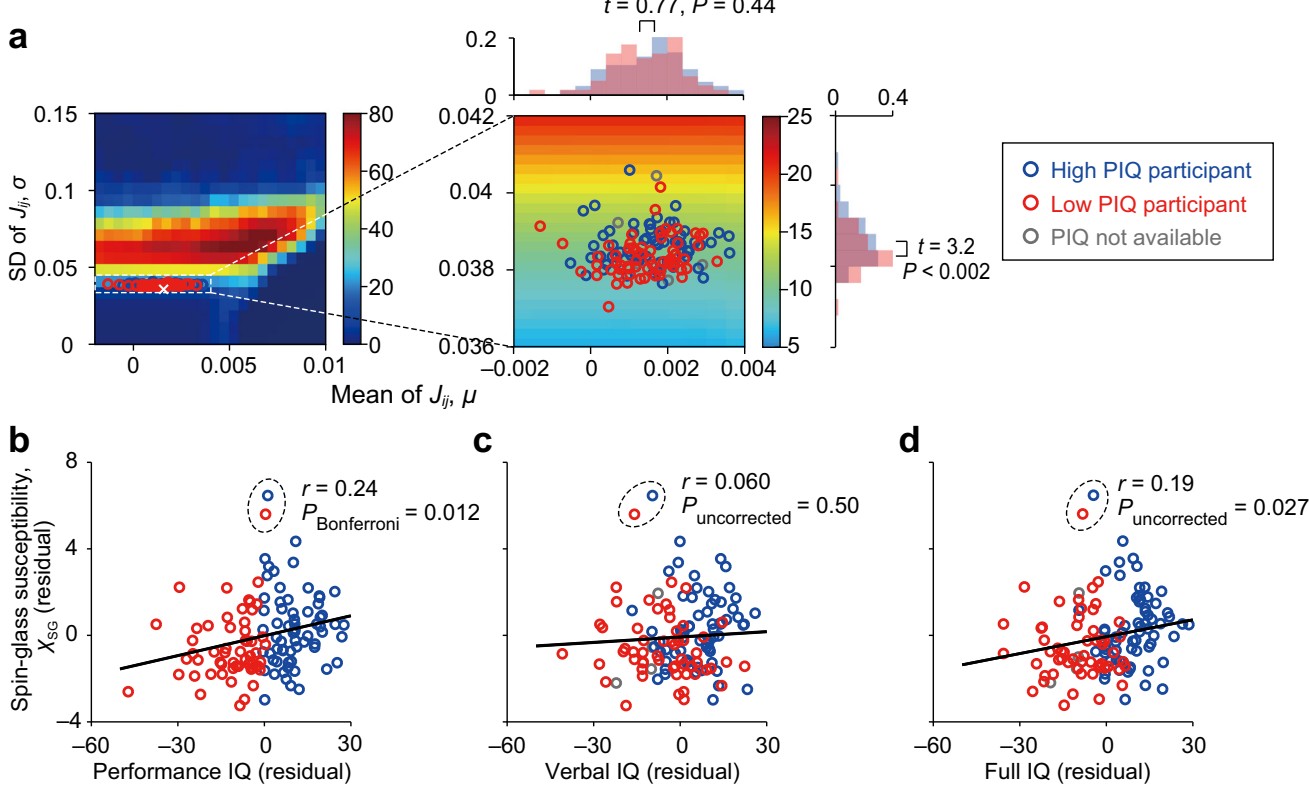

**Fig. 2 Association between the spin-glass susceptibility and the IQ scores. a** Magnification of Fig. 1c. The blue and red circles represent participants with a high performance IQ score (≥109) and a low performance IQ score (<109), respectively. The two overlapping histograms on the horizontal axis are the distributions of $\tilde{\mu}$ for each participant group. The histograms on the vertical axis are the distributions of $\tilde{\sigma}$. **b** Relationship between $\chi_{SG}$ and the performance IQ. A solid circle represents a participant. The participants enclosed by the dashed circle represent outliers determined by Tukey's 1.5 quartile criteria[45]. The Pearson correlation value (i.e., $r$) and the $P$ value shown in the figure are those calculated in the presence of the outliers. The solid line is the linear regression. **c** Relationship between $\chi_{SG}$ and the verbal IQ. **d** Relationship between $\chi_{SG}$ and the full IQ. The $\chi_{SG}$ and IQ values shown in **b**, **c**, and **d** are those after the effects of the age and the gender have been partialed out.

**Consistency with the critical slowing down analysis.** The previous literature used various measures of criticality. We measured for each participant such a measure, i.e., the scaling exponent of autocorrelation[47,48]. This measure quantifies the critical slowing down phenomenon, which has been observed in critical states of the brain[48]. Note that this index quantifies temporal correlation and is orthogonal to what we have measured. We computed the scaling exponent for the autocorrelation function of the fMRI signal at each ROI, using the detrended fluctuation analysis[47,48]. Then, we took the average of the scaling exponent over the $N = 264$ ROIs for each participant, which is denoted by $\alpha$. The association between $\alpha$ and the IQ scores was consistent with the results for $\chi_{SG}$ ($\alpha$ vs performance IQ: $r_{129} = 0.29$, $P_{Bonferroni} = 0.002$; $\alpha$ vs verbal IQ: $r_{126} = 0.19$, $P_{Bonferroni} = 0.068$; $\alpha$ vs full IQ: $r_{130} = 0.25$, $P = 0.003$). These results were robust against the removal of outliers ($\alpha$ vs performance IQ: $r_{128} = 0.28$, $P_{Bonferroni} = 0.002$; $\alpha$ vs verbal IQ: $r_{125} = 0.17$, $P_{Bonferroni} = 0.10$; $\alpha$ vs full IQ: $r_{130} = 0.25$, $P = 0.003$).

We then performed a multivariate linear regression of the performance IQ with $\chi_{SG}$ and $\alpha$ being the independent variable. We found a significant regression equation ($F_{2,128} = 8.0$, $P < 0.001$, adjusted $R^2 = 0.11$). Both $\chi_{SG}$ and $\alpha$ were significantly correlated with the performance IQ ($\chi_{SG}$: $\beta = 0.18$, $P = 0.039$; $\alpha$: $\beta = 0.24$, $P = 0.0067$). This result implies that the association between $\chi_{SG}$ and the performance IQ that we have found is not a byproduct of that between $\alpha$ and the performance IQ. The variance inflation factor for both independent variables was equal to 1.07; this value is small enough for justifying the use of the multivariate regression.

**Effects of data length and individual variability.** We examined if the limited data length and between-participant variability in our data influenced our results. First, we investigated how the estimation of the individual participant's $\chi_{SG}$ and $\chi_{uni}$ depended on the length of her/his fMRI data (Supplementary Fig. 4a). The results were qualitatively the same as those obtained with all the data if we used approximately more than two thirds of the data (i.e., number of volumes per participant larger than ≈ 150). The correlation between $\chi_{SG}$ and the IQ scores and that between $\chi_{uni}$ and the IQ scores were also preserved with the aforementioned data length (Supplementary Fig. 4b, c). Therefore, our main results based on the $\chi_{SG}$ and $\chi_{uni}$ are considered to be reliable in terms of the data length.

Second, as we did in our previous studies[21,37], we divided the participants into two subgroups of the same size and ran some of the main analyses for the subgroups. We started by comparing the pairwise activity correlation, $\langle S_i S_j \rangle$, for each $(i, j)$ pair between the two subgroups. The $\langle S_i S_j \rangle$ values were strongly correlated between the subgroups and also between the empirical data and estimated PMEMs for the two subgroups (Supplementary Fig. 5). We further confirmed that the phase diagrams were similar between the two subgroups (Supplementary Fig. 6). Moreover, we estimated $\tilde{\mu}$ and $\tilde{\sigma}$ for each participant only using the subgroup of participants to which the focal participant belongs. The results were similar to those estimated based on all the participants (Supplementary Fig. 7). Therefore, we conclude that the estimation of the phase diagrams (Fig. 1a–h) and their derivatives (i.e., $\tilde{\mu}$ and $\tilde{\sigma}$), which are based on the estimated phase diagrams,

are robust enough against fluctuations in data, such as those caused by a reduced number of participants.

## Discussion

We provided empirical support that neural dynamics of humans with higher intellectual ability are closer to critical. The present results are consistent with the standing claim of the "critical brain hypothesis" and "edge-of-chaos computation", which jointly dictate that the brain is maximizing its computational performance by poising its dynamics close to the criticality, particularly the criticality involving a chaotic regime.

Here we presented an explicit, albeit only moderate, correlation between the IQ scores and the distance from criticality at an individual's level. Human intelligence has been shown to be associated with genetic factors, brain size, the volume of specific brain regions[49], and the structure of brain networks[26,49]. The present results derived from dynamic fMRI signals provide an orthogonal account of human intelligence as compared with these previous studies and are consistent with the view that cognition is a dynamical process linked to neural dynamics[18,19].

A previous study showed that sleep deprivation pulls the brain dynamics away from the criticality[50]. This result is consistent with ours because sleep deprivation generally compromises one's cognitive and intellectual functions[51].

Previous studies showed that the functional connectivity between particular pairs of ROIs or between subsystems of the brain in the resting state was correlated with intellectual ability[49,52]. These previous results are consistent with ours in the sense that the SG susceptibility can be regarded as the square sum of a type of functional connectivity over the pairs of ROIs and the intellectual score was positively correlated with the SG susceptibility in our analysis. In contrast to these previous studies, which looked at individual connectivity between two regions or subnetworks, we considered $N = 264$ ROIs scattered over the brain[41] as a single functional network. We took this approach for two reasons. First, intelligence is considered to depend on large-scale brain networks[26,52–54]. Second, phase diagram analysis ideally requires a thermodynamic limit, i.e., infinitely many ROIs. One strategy to further approach the thermodynamic limit is to use a single voxel acquired by MRI as a node, significantly scaling up $N$. In this case, spatial correlation among ROIs, which we have ignored in the present study, would be prominent. Because spatial dimensionality affects the phase diagrams even qualitatively[30], this case may require two- or three-dimensional SG models. We leave this as a future problem. The literature also suggest that specific brain systems such as the fronto-parietal network[55] and the default-mode network[56] predict intelligence of humans. Running the same analysis for these and other brain systems to seek specificity of the results warrants future work. Because the present method requires hundreds of ROIs, we may benefit from considering voxel-wise networks of a specific brain system that allow many ROIs for particular brain systems.

In our previous paper, we posed the limited accuracy of fitting the PMEM to fMRI data when $N$ is large[38]. The argument was based on the probability that each of the $2^N$ possible activity patterns appears compared between the empirical data and the estimated PMEM. In the present manuscript, we have not used this accuracy measure, because it cannot be calculated when $N$ is large. Instead, we validated the model by confirming that the difference between the empirical data and estimated PMEM in terms of the signal average, $\langle S_i \rangle$, and the pairwise correlation, $\langle S_i S_j \rangle$, is small (Supplementary Fig. 8). This approach is based on the assumption that the average and second order correlation of signals explain most of the information contained in the given data, which has been confirmed for smaller $N$ in previous studies

using fMRI data[21,24,37,38]. Although only comparing $\langle S_i \rangle$ and $\langle S_i S_j \rangle$ between the data and model is a weaker notion of accuracy of fit than using the accuracy measure[38], the former approach has widely been accepted, explicitly or implicitly, in the literature[57,58]. However, we point out that how to justify the use of PMEMs when $N$ is large remains an open issue.

There are various types of criticality, corresponding to different types of phase transitions. Within the framework of the Ising model, we showed that human fMRI data were in the paramagnetic phase and were close to the boundary with the SG phase but not to the boundary with the ferromagnetic phase. Furthermore, high fluid intelligence was associated with the proximity to the boundary between the paramagnetic and SG phases. In theory, the SG phase yields chaotic dynamics in spin systems including the SK model[31–33], whereas the ferromagnetic phase is obviously non-chaotic. Therefore, although the definition of the chaos in the SG phase is different from that observed in cellular automata[15] and recurrent neural networks[16,17], our results are consistent with the idea of enhanced computational performance at the edge of chaos.

The previous accounts of the critical brain or critical neural circuits are mostly concerned with phase transitions different from the paramagnetic–SG phase transition or its analogs. Examples include phase transitions between quiescent (i.e., subcritical) and active (i.e., supercritical) phases as an excitability control parameter changes[11,12,59–61], between ordered and chaotic phases as connectivity parameters change[17], between a low-activity monostable state and a high-activity multistable state[62], and the divergence of heat capacity[5,6,35,36]. Note that, in the theory of the Ising models, the heat capacity diverges on the boundary between the paramagnetic and ferromagnetic phases, whereas it increases without diverging on the boundary between the paramagnetic and SG phases[30]. Most of these previous results based on the Ising model related neural dynamics to the paramagnetic–ferromagnetic phase transition rather than the paramagnetic–SG transition. Roughly speaking, paramagnetic and ferromagnetic phases correspond to active and quiescent phases, respectively. Computational studies also support the ferromagnetism[13,63,64]. In contrast, we provided a signature of the paramagnetic–SG phase transition, not the paramagnetic–ferromagnetic transition. Fraiman et al. reported that the Ising model at the paramagnetic–ferromagnetic phase transition explains properties of functional networks based on fMRI data[63]. They used a two-dimensional Ising model with a uniform strength of interaction between pairs of nodes that are adjacent on a square lattice (and $J_{ij} = 0$ for the rest of pairs). Another study that suggested the paramagnetic–ferromagnetic phase transition for fMRI signals also assumed a uniform $J_{ij}$[64]. In contrast, we did not constrain the $J_{ij}$ values and instead inferred the $J_{ij}$ values (i.e., structure of functional network) using the PMEM. Because these previous studies[63,64] did not assume heterogeneity in $J_{ij}$ as we did, their results do not contradict ours. In fact, the assumption of a uniform $J_{ij}$ corresponds to setting $\sigma = 0$ in our phase diagrams. If one varies $\mu$ under the condition $\sigma = 0$, the only possible phase transition is the paramagnetic–ferromagnetic transition (Fig. 1a–d). However, that phase transition point, which is derived under the condition $\sigma = 0$, is far from the location of the empirical data when $\sigma$ is allowed to deviate from 0 (crosses in Fig. 1a–d). Therefore, allowing heterogeneity in $J_{ij}$ may be key to further clarifying the nature of critical neural dynamics.

We showed that neural dynamics for each participant were close to but substantially off the criticality separating the paramagnetic and SG phases. Other studies using the PMEM[65] and other models[66] also support off-critical as opposed to critical neural dynamics in the brain. A study applying the PMEM to local field potentials suggested that such off-critical dynamics

may potentially have functional advantages because the off-critical situation would prevent the dynamics to get past the phase boundary to enter the other phase under the presence of noise[66]. The other phase may correspond to pathological neural dynamics such as epilepsy. The off-critical neural dynamics that we found for our participants, regardless of their IQ scores, may benefit from the same functional advantage.

Applying the current analysis pipeline to various neuroimaging and electrophysiological data in different contexts, from health to disease, and during rest and tasks, to evaluate the relevance of the different types of phase transitions warrants future work. For example, as a disease progresses, the brain dynamics may be gradually altered to transit from one phase to another, or to approach or repel from a phase transition curve. In fact, the method is applicable to general multivariate time series. Deployment of the present method to other biological and non-biological data may also be productive.

One could classify the data from participants with high and low IQ scores using a simple multivariate Gaussian decoder[67]. Such a decoder would assume as input the mean and covariance of the fMRI data for each participant or its random samples having the same mean and covariance. In fact, multivariate Gaussian distributions having the same covariance structure as the empirical data yielded similar results (Supplementary Fig. 9). Because our PMEM also assumed the same input but was not optimized for classifying the participants, an optimized Gaussian decoder will probably be more efficient than our PMEM in explaining the IQ scores of the participants. This approach is conceptually much simpler than the present one, which employ the PMEM and its phase diagrams. However, the aim of the present study was to find empirical support of the critical brain hypothesis by relating the fMRI data to the phase diagrams of an archetypal spin system rather than to efficiently classify participants.

We found that the SG susceptibility was positively, although not strongly, correlated with individual differences in the performance IQ score but not in the verbal IQ score. The verbal IQ reflects individuals' knowledge about verbal concepts and crystalized intelligence[43]; crystalized intelligence refers to one's cognitive functioning associated with previously acquired knowledge and skills. In contrast, the performance IQ reflects fluid intelligence, which refers to active or effortful problem solving and maintenance of information[39]. Our results imply that the critical brain dynamics may be particularly useful for active and flexible cognitive functions.

## Methods

**Participants**. One-hundred thirty eight ($n = 138$) healthy and right-handed adult participants (54 females and 84 males) in the Nathan Kline Institute's (NKI) Rockland phase I Sample[68] were analyzed. The data collection was approved by the institutional review board of the Nathan Kline Institute (no. 226781). Written informed consent was obtained for all the participants. Although the data set contains a wide range of the age (18–85 yo), the present results were not an age effect because the IQ values are standardized for age[42] and because we have partialed out the effect of age (and the gender) in the present analysis. Participants' IQ scores were derived from the Wechsler Abbreviated Scale of Intelligence[42]. We used the full scale IQ (full IQ for short), performance IQ, and verbal IQ.

**Preprocessing**. We used the same MRI data and the same preprocessing pipeline as our previous study's[69], except that we used resting-state fMRI signals from 264 ROIs, whose coordinates were derived in the previous literature[41]. In short, we submitted the resting-state fMRI data in the NKI Rockland phase I Sample with TR = 2500 ms and for 10 min 55 s for each participant to our preprocessing pipeline in FSL and applied band-pass temporal filtering (0.01–0.1 Hz).

The obtained fMRI signals $x_i(t)$ ($i = 1, \ldots, N; t = 1, \ldots, t_{max}$, where $t_{max} = 258$) were transformed into their $z$-values using $z_i(t) = (x_i(t) - \mu(x(t)))/\sigma(x(t))$, where $\mu(x(t))$ and $\sigma(x(t))$ represent the average and standard deviation of $x_i(t)$ over the $N$ ROIs, respectively. Note that $\mu(x(t))$ is the global signal. When we tested the robustness of the results by not removing the global signal, we set $z_i(t) = x_i(t)$.

We binarized the signal as follows:

$$S_i(t) = \begin{cases} +1 & \text{if } z_i(t) \geq 0, \\ -1 & \text{if } z_i(t) < 0. \end{cases} \tag{4}$$

**Estimation of h and J by pseudo-likelihood maximization**. The probability of each of the $2^N$ activity patterns is equal to its frequency of occurrence normalized by the $t_{max}$ time points and 138 participants. We fitted the Ising model to this probability distribution on the $2^N$ activity patterns.

We estimated the parameter values of the Ising model (i.e., **h** and **J**) by maximizing a pseudo-likelihood (PL)[38,70]. We approximate the likelihood function by

$$\mathcal{L}(\mathbf{h}, \mathbf{J}) \approx \prod_{t=1}^{t_{max}} \prod_{i=1}^{N} \tilde{P}(S_i | \mathbf{h}, \mathbf{J}, \mathbf{S}_{/i}(t)), \tag{5}$$

where $\tilde{P}$ represents the conditional Boltzmann distribution for a single spin, $S_i \in \{-1, 1\}$, when the $S_j$ values ($j \neq i$) are equal to $\mathbf{S}_{/i}(t) \equiv (S_1(t), \ldots, S_{i-1}(t), S_{i+1}(t), \ldots, S_N(t))$, i.e.,

$$\tilde{P}(S_i | \mathbf{h}, \mathbf{J}, \mathbf{S}_{/i}(t)) = \frac{\exp\left[h_i S_i + \sum_{j=1, j \neq i}^{N} J_{ij} S_i S_j(t)\right]}{\sum_{S_i' = -1, +1} \exp\left[h_i S_i' + \sum_{j=1, j \neq i}^{N} J_{ij} S_i' S_j(t)\right]}. \tag{6}$$

In Eq. (6), one determines the probability of each activity pattern under the assumption that $S_j$ ($j \neq i$) does not change when drawing the value of $S_i$ ($i = 1, \cdots, N$). We ran a gradient ascent updating scheme given by

$$h_i^{new} - h_i^{old} = \epsilon\left(\langle S_i \rangle_{empirical} - \langle \overline{S_i} \rangle_{\tilde{P}}\right), \tag{7}$$

$$J_{ij}^{new} - J_{ij}^{old} = \epsilon\left(\langle S_i S_j \rangle_{empirical} - \langle \overline{S_i S_j} \rangle_{\tilde{P}}\right), \tag{8}$$

where $\langle \overline{S_i} \rangle_{\tilde{P}}$ and $\langle \overline{S_i S_j} \rangle_{\tilde{P}}$ are the mean and correlation with respect to distribution $\tilde{P}$ (Eq. (6)) and given by

$$\langle \overline{S_i} \rangle_{\tilde{P}} = \frac{1}{t_{max}} \sum_{t=1}^{t_{max}} \tanh\left[h_i + \sum_{j'=1, j' \neq i}^{N} J_{ij'} S_{j'}(t)\right] \tag{9}$$

and

$$\langle \overline{S_i S_j} \rangle_{\tilde{P}} = \frac{1}{t_{max}} \sum_{t=1}^{t_{max}} S_j(t) \tanh\left[h_i + \sum_{j'=1, j' \neq i}^{N} J_{ij'} S_{j'}(t)\right], \tag{10}$$

respectively. It should be noted that this updating rule avoids the calculation of $\langle S_i \rangle$ and $\langle S_i S_j \rangle$ with the original spin system, Eqs. (1) and (2), which is computationally formidable with $N = 264$. As $t_{max} \to \infty$, the estimator obtained by the PL maximization approaches the exact maximum likelihood estimator[70]. In fact, the Ising model with the estimated parameter values $\mathbf{h} = \hat{\mathbf{h}}$ and $\mathbf{J} = \hat{\mathbf{J}}$ produced the mean and correlation of spins in the empirical data with a sufficiently high accuracy (Supplementary Fig. 8).

We previously provided MATLAB code for estimating the Ising model from data by PL maximization[38]. The code is publicly available on GitHub repository (https://github.com/tkEzaki/energy-landscape-analysis).

**Monte Carlo simulation**. We used the Metropolis method[71] to calculate the observables of the Ising model estimated from the empirical data and the SK model. In each time step, a spin $S_i$ was chosen uniformly at random for being updated. The selected spin was flipped with probability $\min\{e^{-\Delta E}, 1\}$, where $\Delta E = E(\mathbf{S}_{flipped}) - E(\mathbf{S})$, **S** is the current spin configuration, and $\mathbf{S}_{flipped}$ is the spin configuration after $S_i$ is flipped. The initial condition was given by $S_i = 1$ with probability 1/2 (and hence $S_i = -1$ with probability 1/2), independently for different $i$'s. We recorded the spin configuration **S** every $N$ time steps.

For the empirical data, we discarded the first $10^6 \times N$ time steps as transient and then recorded $10^7$ samples of **S** in total. Based on the $10^7$ samples, we calculated the averages of the observables (i.e., $|m|$, $q$, $\chi_{SG}$, $\chi_{uni}$, and $C$). For drawing the phase diagrams with the $N = 264$ ROIs, we further averaged each observable over ten independent simulations starting from different initial spin configurations. In Fig. 1k, we averaged the $\chi_{SG}$ value over 40 combinations of $N'$ ROIs out of the 264 ROIs as well as over $10^7$ samples and ten initial conditions.

For the phase diagram for the SK model, we discarded the first $10^4 \times N$ time steps as transient and then collected $5 \times 10^4$ samples of **S** from each of $10^3$ realizations of **J**. We drew the phase diagrams on the basis of the $5 \times 10^4 \times 10^3 = 5 \times 10^7$ samples.

**Estimation of μ and σ for single participants**. The estimation of the empirical interaction matrix, $\hat{\mathbf{J}}$, requires a large amount of data, or practically, concatenation of fMRI data across different participants. Therefore, one cannot directly compute the mean and standard deviation of $\hat{\mathbf{J}}$ (i.e., $\mu$ and $\sigma$) for each participant. Given this constraint, we estimated $\mu$ and $\sigma$ for each participant (denoted by $\tilde{\mu}$ and $\tilde{\sigma}$) using

the $\chi_{SG}$ and $\chi_{uni}$ values for the participant (denoted by $\tilde{\chi}_{SG}$ and $\tilde{\chi}_{uni}$) as follows (Supplementary Fig. 10).

First, we examined the phase diagrams in terms of $\chi_{SG}$ and $\chi_{uni}$ generated for the collection of all participants (Fig. 1c, d). Specifically, we calculated $\chi_{SG}(\mu, \sigma)$ and $\chi_{uni}(\mu, \sigma)$ values at $\mu = \mu_k$ ($k = 1, \ldots, 25$), where $\mu_1 = -0.002$, $\mu_2 = -0.0015$, $\ldots$, $\mu_{25} = 0.01$, and $\sigma = \sigma_\ell$ ($\ell = 1, \ldots, 21$), where $\sigma_1 = 0$, $\sigma_2 = 0.0075$, $\ldots$, $\sigma_{21} = 0.15$.

Second, at each $\mu_k (k = 1, \ldots, 25)$, we computed the value of $\check{\sigma}_k$ satisfying $\chi_{SG}(\mu_k, \check{\sigma}_k) = \tilde{\chi}_{SG}$ (Supplementary Fig. 10a, c) using a linear interpolation of $\chi_{SG}(\mu_k, \sigma_\ell)$ ($\ell = 1, \ldots, 21$), i.e., $\check{\sigma}_k = \alpha\sigma_{\ell'} + (1 - \alpha)\sigma_{\ell'+1}$, where $\ell'$ ($1 \leq \ell' \leq 21$) is the integer satisfying $\chi_{SG}(\mu_k, \sigma_{\ell'}) \leq \tilde{\chi}_{SG} < \chi_{SG}(\mu_k, \sigma_{\ell'+1})$, and $\alpha = [\chi_{SG}(\mu_k, \sigma_{\ell'+1}) - \tilde{\chi}_{SG}]/[\chi_{SG}(\mu_k, \sigma_{\ell'+1}) - \chi_{SG}(\mu_k, \sigma_{\ell'})]$. Because $\chi_{SG}(\mu_k, \sigma_\ell)$ increases with $\ell$ in the paramagnetic phase, the $\ell'$ value is uniquely determined for each $k$, if it exists. In this manner, we obtained a piecewise linear curve whose knots were $(\mu_k, \check{\sigma}_k)$ ($k = 1, \ldots, 25$). On this curve, $\chi_{SG}(\mu, \sigma)$ is approximately equal to $\tilde{\chi}_{SG}$ (Supplementary Fig. 10e, g). It should be noted that we have assumed that $(\tilde{\mu}, \tilde{\sigma})$ to be estimated is near $(\hat{\mu}, \hat{\sigma})$ computed for the entire population (represented by the cross in Fig. 1a–d). More precisely, we are searching $(\tilde{\mu}, \tilde{\sigma})$ in the vicinity of the paramagnetic–SG phase boundary on the paramagnetic side. This assumption is supported by the empirical values of $m$ and $q$ for individual participants, i.e., $m = -8.0 \times 10^{-3} \pm 7.8 \times 10^{-3}$ (mean $\pm$ SD) and $q = 3.4 \times 10^{-3} \pm 0.4 \times 10^{-3}$.

Third, we calculated a piecewise linear curve on which $\chi_{uni}(\mu, \sigma)$ was approximately equal to $\tilde{\chi}_{uni}$ (Supplementary Fig. 10f, g). To this end, we applied the same algorithm as the one used in the previous step but by fixing $\sigma_\ell$ (Supplementary Fig. 10b) and finding $\check{\mu}_\ell$ (Supplementary Fig. 10d), exploiting the fact that $\chi_{uni}(\mu_k, \sigma_\ell)$ monotonically increases with $\mu$ in the paramagnetic phase.

Finally, we computed $(\tilde{\mu}, \tilde{\sigma})$ for the individual as the intersection of the two piecewise linear curves (Supplementary Fig. 10g).

**Specific heat.** The specific heat is defined by

$$C = \frac{\langle E^2 \rangle - \langle E \rangle^2}{NT^2}, \qquad (11)$$

where $T$ is the temperature. We set $T = 1$ because we implicitly did so in Eqs. (1) and (2).

To compute $C$ for each participant, we first drew a phase diagram for $C$ in terms of $\mu$ and $\sigma$ for the entire population (Supplementary Fig. 11a). The obtained phase diagram was similar to that for the SK model (Supplementary Fig. 11b). Then, we determined the $C$ value for each participant as the point in the phase diagram corresponding to the $(\mu, \sigma)$ for the participant. Because the phase diagram for $C$ is drawn for discrete values of $\mu$ and $\sigma$, we applied the standard bilinear interpolation to determine the $C$ value corresponding to a given $(\mu, \sigma)$.

**Statistics and reproducibility.** Statistical tests were performed using SPSS 24.0. The details of each analysis are found in prior sections.

**Reporting summary.** Further information on research design is available in the Nature Research Reporting Summary linked to this article.

## Data availability
The data set used in this study (Nathan Kline Institute Rockland phase I Sample) is publicly available (http://fcon_1000.projects.nitrc.org/indi/pro/nki.html).

## Code availability
The code used in this study is available upon request.

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

## Acknowledgements

The authors would like to thank Yoshiyuki Kabashima for useful discussions. T.E. acknowledges the support provided through PRESTO, JST (No. JPMJPR16D2) and Kawarabayashi Large Graph Project, ERATO, JST. M.S. acknowledges the support provided through the European Commission (CIG618600) and JSPS (16H02053; 16H05959; 16H06406; 16KT0002). T.W. acknowledges supports from JSPS (18H06094), Yamaha Sports Challenge Fellowship, Fukuhara Fund for Applied Psychoeducation Research, and SENSHIN Medical Research Foundation. N.M. acknowledges the support provided through, CREST, JST (No. JPMJCR1304) and Kawarabayashi Large Graph Project, ERATO, JST, and EPSRC Institutional Sponsorship to the University of Bristol.

## Author contributions

N.M. designed research; T.E., E.F.R. and N.M. contributed new analytic tools; T.E. and M.S. analyzed data; T.E., E.F.R., T.W., M.S. and N.M. performed research; T.E., T.W., M.S. and N.M. wrote the paper.

## Competing interests

The authors declare no competing interests.
