## [Peer Review File · Communications Biology]

Reviewers' comments:

Reviewer #1 (Remarks to the Author):

Examination of the paramagnetic and ferromagnetic border in the Ising model was performed using simulations followed by a comparison with resting fMRI data from ~150 participants. The authors show that in the sigma/mu plane, individual participant's dynamical signatures cluster close to the paramagnetic border, which the authors state to be equivalent to an 'edge-of-chaos' border. Fluid intelligence scores based on the Wechsler Intelligence Scale correlated with closeness to the paramagnetic border. Results suggest that the ability to think logically and find solutions, but not crystallized intelligence (prior knowledge, experience and verbal expression) improves with the brain resting closer to criticality.

Overall impression of the work

Results seem robust and controls are provided that include threshold independence and robustness to removal of z-score normalization. Manuscript is reasonably well written (particularly Results).

I have a number of comments that might improve the manuscript in its present state:

Introduction:

Please reexamine the use of your references. When it comes to avalanches and criticality the main work is Beggs and Plenz, 2003. In depth reviews with respect to brain dynamics and criticality besides Chialvo Nat Phys 2010 would be Plenz The European Physical Journal 2012.

Second sentence in Introduction doesn't make sense " ... including criticisms such as the lack of power laws in the relevant observables." This seems a bit off – it is more that identifying a power law is a necessary but not sufficient condition for critical dynamics. Please elaborate and reword.

Langton 1994 could never prove that there is a second order phase transition in his simulations. The wording by the authors suggests otherwise. Please clarify.

"According to the critical brain hypothesis ..." – better 'One prediction from the critical brain hypothesis ..."

"However, neuronal avalanches do not imply transitory dynamics or their absence." This sentence is difficult to understand. The argument is not clear at all. Please elaborate.

" chaotic dynamics ... from ... healthy controls more strongly than ..." The construction of this argument and how it fits into the authors' logic is not clear. The whole paragraph needs profound reworking to make the authors' arguments more clear.

" ... different degrees of intelligence.' Consider rewording like ' correlate with IQ scores'.

Discussion: Needs rewording in many places. ' ... more intelligent human brains' – needs rewording. "The criticality view of the brain is not new.' Not sure what this sentence states beyond the obvious and not having any references doesn't help either. I recommend a native English speaker to comb through the discussion for rewording some of the most problematic statements regarding human intelligence.

Consider Meisel et al 2017 on critical slowing down changes in humans with wakefulness as this could pose some limits on measuring critical slowing down in humans as done by the authors in this study.

Reviewer #2 (Remarks to the Author):

This article fits maximum entropy models to binarized fMRI data, determines the proximity of each participant to a phase transition, and shows a correlation with the IQ across all subjects.

The result is interesting. I would have expected IQ variability to be very subtle and difficult to find significant correlations with metrics from statistical physics where reports of changes across more drastically different brain states (e.g. anesthesia, sleep, coma) are yet to be published.

My main criticism concerns the fact that these results are not informative in terms of the underlying neurobiology; the authors found correlations with observables extracted from whole brain imaging data, but I would expect that certain regions and circuit are more involved than others in fluid intelligence. Perhaps the authors could restrict their analysis to a subset of the regions of interest and try to find which regions are necessary for the reported correlation.

My other criticism is that of presentation; I consider that the paper is difficult to read for non physicists. The methods section could be more didactic in this sense (perhaps extended in the supplementary information). Different ansatz could be more properly motivated, and worked examples provided as well.

Reviewer #3 (Remarks to the Author):

This paper adds to the large body of evidence that the brain operates near a critical point. The authors build on their recent work on fitting the Ising model to functional neuroimaging data. They find they can map subjects in a phase diagram and identify the type of phase transition nearby. They also find a moderate correlation between a measure of distance to criticality and a measure of IQ, suggesting some functional relevance to their model findings. Given the high level of interest in criticality in the brain this paper should find a good audience. The links to IQ are not particularly strong (and the authors are appropriately circumspect on this), but nevertheless this is an intriguing finding that fits into the broader narrative on functional benefits of the near-critical regime. However, I do have some methodological concerns, and some of the writing is not really geared for a biology audience. The paper would be improved if the authors address the following:

1. My biggest concern is over the fitting of the Ising model. From my understanding of these methods, partly from an earlier paper by the authors (Ezaki et al. 2017, Ref. 37 here), data length is a big problem when the number of ROIs is high. With 264 ROIs there are 2^{264} different states so the probability distribution over these states will necessarily be extremely sparsely sampled. In that earlier paper the guidance was that accuracy of the fit scales as a function of $t_{\max}/2^N$, so that for typical fMRI values of t_{\max} one can only look at $N \sim 5$ accurately, or $N \sim 8$ if pooling over 10 subjects (and perhaps $N \sim 12$ with the 138 subjects here?). What is new here to overcome the earlier paper's advice that "Currently we cannot apply the method to relatively large brain systems (i.e. those with a larger number of ROIs)"? This would seem to be a big advance.

2. If I understand correctly, the single-subject estimation of position in the phase diagram proceeds by first calculating the phase diagram for the group concatenated data, then estimating each subject's σ and μ via interpolation using empirically-calculated χ_{SG} and χ_{uni} . These latter quantities are presumably somewhat noisily estimated given the short data length (how exactly are

they "calculated for each individual only from the covariance matrix of the data"?). In light of the potential inaccuracy of the fitting (point 1 above) and fact that the IQ correlation is found for a relatively narrow range of sigma where small shifts could change ordering, it seems plausible that the individual subjects may not necessarily behave the same as the group-level phase diagram. Is it possible to estimate the uncertainty in the group-level phase diagram (e.g. via a nonparametric bootstrap or leave-one-out method or similar), and propagate this forward to the values used in the IQ correlation? I worry that estimation errors could affect the robustness of the results.

3. There are some pieces of physics taken for granted here that should probably be explained for a biology journal, especially on p2-3 with lots of cites to a book Ref. 29. Should explain notions of paramagnetic and ferromagnetic phases, what exactly the SK model is, how large sigma corresponds to low temperature, what is meant by "transitory dynamics" and how this relates to criticality, what it means to not find a phase transition between the paramagnetic and ferromagnetic phases, and the discussion text on chaos in different regimes.

4. p4, 2nd para, bit incongruous to refer to $r=0.21$ as "mildly correlated" versus $r=0.24$ on the previous page being referred to as "significantly positively correlated".

5. p5, the discussion of how these results are consistent with balanced excitation and inhibition is unclear and unconvincing.

6. p5, re discrepancy with studies finding ferromagnetism in fMRI data, the explanation given in terms of the present research using fMRI data seems like faulty logic (same type of data no?) How do the results reconcile with those of e.g. Ref. 55?

Reviewer #4 (Remarks to the Author):

In the present article, Ezaki et al. used a maximum entropy model to map the whole-brain BOLD dynamics onto a phase diagram. They showed that dynamics were poised in the paramagnetic phase and close to a spin-glass phase transition. They then showed that subjects with larger performance IQ presented dynamics that were closer to that phase transition.

I enjoyed reading the article, which is clearly written. The method presented therein will be very useful for the neuroscience community and, thus, I think that the code to learn the model should be made publicly available. However, I have some concerns that should be addressed in order to improve the article:

1) My first concern relates to the binarization of the fMRI data. The authors binarized the data by first using z-score and setting a value equal to 1 if fluctuations were positive and -1 otherwise. Thus, by construction of the binary data, the magnetization must be $m = 0$, forcing the system to be in the paramagnetic phase. I think that taking the standard deviation as the threshold (as it is the common practice) would imply $m > 0$. In figure S2, the authors varied the binarization threshold, but they only showed the correlation between the IQ and spin-glass susceptibility (which depends on the covariance of the data). The authors should show how the point representing the data in the phase diagram changes with different thresholds.

2) Also, it would be interesting to compare the results with those from surrogate data. Specifically, the authors could generate multivariate gaussian time-series with the same covariance as the empirical

data, apply the threshold, and see where the surrogate data are placed on the phase diagram. My guess is that the surrogates would be placed close to the spin-glass transition. Moreover, the data from subjects with large and low IQ could be classified using a simple Gaussian decoder, based on the covariance of the data. If this simple decoder achieves better discrimination than using the model parameter σ , the results would be weakened. In other words, if a simple model such as a multivariate Gaussian process leads to better discrimination and relatively same working point in the phase diagram, then the proposed model is somehow redundant (and conceptually more complex).

3) The maximum entropy model was built using the binary data of all subjects. Because the data could be highly variable between subjects, the model could be biased to replicate this diversity. To test this, the authors should learn two models using the halves of the data, test the model built from the first-half data on the second-half data and vice versa, and evaluate the likelihood ratios (train vs test). Likelihood ratios close to 1 would indicate that the model can generalize.

4) The mapping of the data in the phase diagram shows rather that the system is subcritical. Recent studies showed that subcritical dynamics are more likely during rest (Priesmann et al., 2014; Hahn et al., 2017), with potential functional advantages. This should be discussed. Note also that, in the finite size scaling analysis, the susceptibility grows with N' but there is no sign of divergence.

5) I am not familiar with IQ studies, but I think that sentences such as "more intelligent human individuals" or "more intelligent human brains" are not appropriate and should be avoid.

References:

V. Priesemann, et al., "Spike avalanches in vivo suggest a driven, slightly subcritical brain state," *Front. Syst. Neurosci.*, 2014.

Hahn G et al. (2017) Spontaneous cortical activity is transiently poised close to criticality. *PLoS Comput Biol* 13(5): e1005543.

I hope the authors find this comments useful.

Responses to Reviewer #1

We appreciate the reviewer's useful suggestions. We have addressed all the comments by the reviewer. The page numbers in our responses refer to those in the revised manuscript unless otherwise mentioned. The deleted and added text is shown in **red** and **blue**, respectively, in the revised manuscript.

Examination of the paramagnetic and ferromagnetic border in the Ising model was performed using simulations followed by a comparison with resting fMRI data from ~150 participants. The authors show that in the sigma/mu plane, individual participant's dynamical signatures cluster close to the paramagnetic border, which the authors state to be equivalent to an 'edge-of-chaos' border. Fluid intelligence scores based on the Wechsler Intelligence Scale correlated with closeness to the paramagnetic border. Results suggest that the ability to think logically and find solutions, but not crystallized intelligence (prior knowledge, experience and verbal expression) improves with the brain resting closer to criticality.

Overall impression of the work

Results seem robust and controls are provided that include threshold independence and robustness to removal of z-score normalization. Manuscript is reasonably well written (particularly Results).

We are glad to hear the overall positive evaluation by the reviewer.

I have a number of comments that might improve the manuscript in its present state:

Introduction:

Please reexamine the use of your references. When it comes to avalanches and criticality the main work is Beggs and Plenz, 2003. Indepth reviews with respect to brain dynamics and criticality besides Chialvo Nat Phys 2010 would be Plenz The European Physical Journal 2012.

We had already cited Beggs and Plenz (2003) and Chialvo (2010) in the previous version. We additionally cited Plenz (2012) in the first sentence of Introduction.

Second sentence in Introduction doesn't make sense " ... including criticisms such as the lack of power laws in the relevant observables." This seems a bit off – it is more that identifying a power law is a necessary but not sufficient condition for critical dynamics. Please elaborate and reword.

We revised this part to read as follows:

"This hypothesis has been investigated for more than two decades including criticisms such as the presence of alternative mechanisms explaining power law scaling in the relevant observables [Touboul2010, Botcharova2012, Hesse2014, Markovic2014]. Experimental evidence such as the recovery of critical behavior after interventions, which is difficult to explain by alternative mechanisms, lends supports to the hypothesis [Hesse2014]."

Langton 1994 could never prove that there is a second order phase transition in his simulations. The wording by the authors suggests otherwise. Please clarify.

In the previous version, we cited Langton 1990 (not 1994) in the second paragraph of Introduction. Therefore, we assume that the reviewer points to this paragraph. The sentence that cited Langton 1990 read "These findings align with the idea of edge-of-chaos computation, with which computational ability of a system is maximized at criticality separating a chaotic phase and a non-chaotic phase [Langton1990, Bertschinger2004, Legenstein2007]". The reviewer may have considered that the word "criticality" connotes second-order transitions. Therefore, we replaced it by **"phase transitions"**.

"According to the critical brain hypothesis ..." – better 'One prediction from the critical brain hypothesis ..."

We adopted the text suggested by the reviewer. Thanks.

"However, neuronal avalanches do not imply transitory dynamics or their absence." This sentence is difficult to understand. The argument is not clear at all. Please elaborate.

We rewrote this sentence as follows:

“Neuronal avalanches are bursts of cascading activity of neurons, whose power-law properties have been related to criticality. However, studies of neuronal avalanches have focused on their scale-free dynamics in space and time, with which statistics of avalanches obey power laws. Scale-free dynamics of neuronal avalanches is a question orthogonal to patterns of transitions between discrete states.”

“... chaotic dynamics ... from ... healthy controls more strongly than ...” The construction of this argument and how it fits into the authors’ logic is not clear. The whole paragraph needs profound reworking to make the authors’ arguments more clear.

We agree. The previous version was unclear on the hypothesis set out in the previous paragraph (i.e., whether or not the relevance of network interaction was part of the hypothesis). Therefore, we extensively revised this and the previous paragraphs. Specifically, we clearly stated that the network interaction was part of the hypothesis. Furthermore, we declined chaotic time series analysis methods due to its irrelevance to network interaction, without negating its potential to be able to extract state transitory dynamics (for non-interacting, single time series). The revised part (i.e., from middle in the previous paragraph to the end of the present paragraph) reads as follows:

“Furthermore, these and other studies [Calhoun2014, Kopell2014, Rabinovich2015] support that state-transition dynamics in the brain involve large-scale brain networks. These arguments are consistent with the proposal that many cognitive functions seem to depend on network connectivity among various regions scattered over the whole brain [Barbey2018]. On these grounds, in the present study we hypothesize that complex and transitory neural dynamics of the brain network (i.e., dynamic transitions among discrete brain states) that are close to criticality are associated with high cognitive performance of humans. There are two major conventional methods for examining criticality and edge-of-chaos computation in empirical neural data. However, they do not correspond to the present hypothesis for their own reasons. First, many of the experimental studies testing the critical brain hypothesis have examined neuronal avalanches [Beggs2003, Beggs2008], including the case of humans [Yu2013, Tagliazucchi2012, Shriki2013]. Neuronal avalanches are bursts of cascading activity of neurons, whose power-law properties have

been related to criticality. However, studies of neuronal avalanches have focused on their scale-free dynamics in space and time, with which statistics of avalanches obey power laws. Scale-free dynamics of neuronal avalanches is a question orthogonal to patterns of transitions between discrete states. Second, nonlinear time series analysis has found that electroencephalography (EEG) signals recorded from the brains of healthy controls are chaotic and that the degree of chaoticity is stronger for healthy controls than individuals with, for example, epilepsy, Alzheimer's disease, and schizophrenia [Stam2005]. However, this method is not usually for interacting time series. Therefore, it does not directly reveal how different brain regions interact or whether possible critical or chaotic dynamics are an outcome of the dynamics at a single region or interaction among different regions."

"... different degrees of intelligence.' Consider rewording like 'correlate with IQ scores'.

We replaced "different degrees of intelligence" by "different intelligence quotient (IQ) scores".

Discussion: Needs rewording in many places. '... more intelligent human brains' – needs rewording. "The criticality view of the brain is not new.' Not sure what this sentence states beyond the obvious and not having any references doesn't help either. I recommend a native English speaker to comb through the discussion for rewording some of the most problematic statements regarding human intelligence.

First, we replaced "more intelligent human brains" by "neural dynamics of humans with higher intellectual ability". Second, we just deleted the sentence "The criticality view of the brain itself is not new." because it is unnecessary and overlaps with the preceding paragraph. Third, we combed through the Discussion section regarding the usage of the terms related to human intelligence (and we do not believe that is the problem of English) and revised the text as follows:

- Second line in the fourth paragraph: "intelligent performance" -> "intellectual ability"
- Fourth line in the fourth paragraph: "the intelligence" -> "the intellectual score"

Consider Meisel et al 2017 on critical slowing down changes in humans with wakefulness as this could pose some limits on measuring critical slowing down in humans as done by the authors in this study.

Because approximate criticality was sustained for about 12 hours since the participants woke up in that study, we consider that their results generally support the brain critical hypothesis except when sleep is deprived. Furthermore, their main result is that sleep deprivation pulls the brain dynamics away from the criticality. This result is in fact consistent with our results because sleep deprivation is expected to make it difficult for participants to exercise cognitive functions at a normal level. We added the following short paragraph to the Discussion section to mention this point and cited the reference (p. 5, lines 215-216).

“A previous study showed that sleep deprivation pulls the brain dynamics away from the criticality [Meisel2017]. This result is consistent with ours because sleep deprivation generally compromises one’s cognitive and intellectual functions [Horne1988].”

Responses to Reviewer #2

We appreciate the reviewer's useful suggestions. We have addressed all the comments by the reviewer. The page numbers in our responses refer to those in the revised manuscript unless otherwise mentioned. The deleted and added text is shown in **red** and **blue**, respectively, in the revised manuscript.

This article fits maximum entropy models to binarized fMRI data, determines the proximity of each participant to a phase transition, and shows a correlation with the IQ across all subjects.

The result is interesting. I would have expected IQ variability to be very subtle and difficult to find significant correlations with metrics from statistical physics where reports of changes across more drastically different brain states (e.g. anesthesia, sleep, coma) are yet to be published.

We are glad to hear positive interests of the reviewer.

My main criticism concerns the fact that these results are not informative in terms of the underlying neurobiology; the authors found correlations with observables extracted from whole brain imaging data, but I would expect that certain regions and circuit are more involved than others in fluid intelligence. Perhaps the authors could restrict their analysis to a subset of the regions of interest and try to find which regions are necessary for the reported correlation.

Thanks for a valuable suggestion. We added the following text to the Discussion section to discuss this issue.

“The literature also suggest that specific brain systems such as the fronto-parietal network [Finn2015] and the default-mode network [Song2009] predict intelligence of humans. Running the same analysis for these and other brain systems to seek specificity of the results warrants future work. Because the present method requires hundreds of ROIs, we may benefit from considering voxel-wise networks of a specific brain system that allow many ROIs for particular brain systems.”

My other criticism is that of presentation; I consider that the paper is difficult to read for non physicists. The methods section could be more didactic in this sense (perhaps extended in the supplementary information). Different ansatz could be more properly motivated, and worked examples provided as well.

We added the following text after Eq. (6) to explain the ansatz underlying the pseudo-likelihood estimation.

“In Eq. (6), one determines the probability of each activity pattern under the assumption that $S_j (j \neq i)$ does not change when drawing the value of $S_i (i = 1, \dots, N)$.”

We also expanded the first subsection of the Results section as follows to supply more pedagogical explanation for non-physicist readers such as biologists:

- First paragraph in the Results section: We removed the mentioning to the Hamiltonian, as it is not necessary and would confuse biology readers who may not know what the Hamiltonian is.
- First paragraph in the Results section: We replaced “To fit the model,” by “Because the model assumes binary data,”.
- We added the following text in the sentence right after equation (2).

“Although we refer to E as the energy, E does not represent the physical energy of a neural system but is a mathematical construct representing the frequency with which activity pattern \mathbf{S} appears in the given data. Activity pattern \mathbf{S} appears rarely in the data if E corresponding to \mathbf{S} is large and vice versa. Parameter h_i represents the tendency that $S_i = 1$ is taken because a positive large value of h_i implies that $S_i = 1$ as opposed to $S_i = -1$ lowers the energy and hence raises the probability that \mathbf{S} with $S_i = 1$ appears. Parameter J_{ij} represents a functional connectivity between ROIs i and j because, if J_{ij} is away from 0, S_i and S_j would be correlated in general.”
- Right after Eq. (3): We supplied the definition of the SK model.
- Right after Eq. (3): We removed a sentence mentioning the relationship between σ and the temperature, because it is unnecessary for readers to understand this connection and because it was slightly inaccurate in its original form due to the μ term in Eq. (3).
- First half of the third paragraph in the Results section: We added text to enhance the explanation of m , q , and each of the three phases.

- Third paragraph in Introduction: To explain what is meant by transitory dynamics, we added "(i.e., dynamic transitions among discrete brain states)".

Because we expanded this part for non-physicists, we did not further modify the Methods sections such as to create worked examples or additional SI text.

Responses to Reviewer #3

We appreciate the reviewer's useful suggestions. We have addressed all the comments by the reviewer. The page numbers in our responses refer to those in the revised manuscript unless otherwise mentioned. The deleted and added text is shown in **red** and **blue**, respectively, in the revised manuscript.

This paper adds to the large body of evidence that the brain operates near a critical point. The authors build on their recent work on fitting the Ising model to functional neuroimaging data. They find they can map subjects in a phase diagram and identify the type of phase transition nearby. They also find a moderate correlation between a measure of distance to criticality and a measure of IQ, suggesting some functional relevance to their model findings. Given the high level of interest in criticality in the brain this paper should find a good audience. The links to IQ are not particularly strong (and the authors are appropriately circumspect on this), but nevertheless this is an intriguing finding that fits into the broader narrative on functional benefits of the near-critical regime. However, I do have some methodological concerns, and some of the writing is not really geared for a biology audience. The paper would be improved if the authors address the following:

We are glad to hear the overall positive evaluation by the reviewer. We addressed the reviewer's concerns one by one as follows.

1. My biggest concern is over the fitting of the Ising model. From my understanding of these methods, partly from an earlier paper by the authors (Ezaki et al. 2017, Ref. 37 here), data length is a big problem when the number of ROIs is high. With 264 ROIs there are 2^{264} different states so the probability distribution over these states will necessarily be extremely sparsely sampled. In that earlier paper the guidance was that accuracy of the fit scales as a function of $t_{\max}/2^N$, so that for typical fMRI values of t_{\max} one can only look at $N \sim 5$ accurately, or $N \sim 8$ if pooling over 10 subjects (and perhaps $N \sim 12$ with the 138 subjects here?). What is new here to overcome the earlier paper's advice that "Currently we cannot apply the method to relatively large brain systems (i.e. those with a larger number of ROIs)"? This would seem to be a big advance.

The reviewer is correct in pointing this out. We added the following paragraph to explain this point.

“In our previous paper, we posed the limited accuracy of fitting the PMEM to fMRI data when N is large [Ezaki2017]. The argument was based on the probability that each of the 2^N possible activity patterns appears compared between the empirical data and the estimated PMEM. In the present manuscript, we have not used this accuracy measure, because it cannot be calculated when N is large. Instead, we validated the model by confirming that the difference between the empirical data and estimated PMEM in terms of the signal average, $\langle S_i \rangle$, and the pairwise correlation, $\langle S_i S_j \rangle$, is small (Supplementary Fig. 8). This approach is based on the assumption that the average and second order correlation of signals explain most of the information contained in the given data, which has been confirmed for smaller N in previous studies using fMRI data [Watanabe2013, Watanabe2014, Ezaki2017, Ezaki2018]. Although only comparing $\langle S_i \rangle$ and $\langle S_i S_j \rangle$ between the data and model is a weaker notion of accuracy of fit than using the accuracy measure [Ezaki2017], the former approach has widely been accepted, explicitly or implicitly, in the literature [Chen2014, Tkacik2014]. However, we point out that how to justify the use of PMEMs when N is large remains an open issue.”

2. If I understand correctly, the single-subject estimation of position in the phase diagram proceeds by first calculating the phase diagram for the group concatenated data, then estimating each subject's sigma and mu via interpolation using empirically-calculated χ_{SG} and χ_{uni} . These latter quantities are presumably somewhat noisily estimated given the short data length (how exactly are they "calculated for each individual only from the covariance matrix of the data"?). In light of the potential inaccuracy of the fitting (point 1 above) and fact that the IQ correlation is found for a relatively narrow range of sigma where small shifts could change ordering, it seems plausible that the individual subjects may not necessarily behave the same as the group-level phase diagram. Is it possible to estimate the uncertainty in the group-level phase diagram (e.g. via a nonparametric bootstrap or leave-one-out method or similar), and propagate this forward to the values used in the IQ correlation? I worry that estimation errors could affect the robustness of the results.

The reviewer's understanding is correct, and we agree with potential limitations of the present study due to these factors. We figure that the reviewer worries about the uncertainty of the result both at the individual participant's level and the group level. Both are due to the short data length. To address these issues, we did the following.

First, we investigated how the estimation of the individual participant's χ_{SG} , χ_{uni} , and the correlation of each with the IQ scores depended on the length of each participant's fMRI data (Fig. S4). We found that the results were qualitatively the same as those obtained with the full length of the data when we used approximately more than two thirds of the data (i.e., data length larger than ~ 150). Therefore, we do not consider that the results based on the χ_{SG} and χ_{uni} values estimated with the present data are too sensitive to noise.

Second, as we did in our previous studies (Refs. [21, 37] in the revised manuscript), we divided the group data into the halves and measured the covariance, $\langle S_i S_j \rangle$ for each (i, j) pair for each half to be compared between each other. The covariance obtained from the two halves was strongly correlated with each other (Fig. S5). We further estimated the PMEM and drew the phase diagram for each half. The phase diagrams were similar to each other (Fig. S6).

Third, in fact, the uncertainty in the group-level phase diagram does not propagate forward to the values used in the IQ correlation. This is because the values used in the IQ correlation are χ_{SG} and χ_{uni} , which are directly calculated from the correlation matrices. However, to wipe out the reviewer's worry, we estimated $\tilde{\mu}$ and $\tilde{\sigma}$ for the individual participants, the calculation of which does need phase diagrams, for the phase diagrams calculated only from a half of the participants. Then, we compared $\tilde{\mu}$ estimated from the half of the participants and $\tilde{\mu}$ estimated from all the participants. We did the same for $\tilde{\sigma}$. The results shown in the new Figure S7 support the robustness of our method regarding the estimation of $\tilde{\mu}$ and $\tilde{\sigma}$ when we used only half the participants.

Therefore, we conclude that the group-level estimation of the phase diagram and the calculation of the individual participants' μ and σ , which is based on the estimated phase diagram, are robust enough against noise, given the current length of the data. We added a paragraph in the main text to describe these results, referring to the SI for more details (p. 5, lines 189-204).

3. There are some pieces of physics taken for granted here that should probably be explained for a biology journal, especially on p2-3 with lots of cites to a book Ref. 29. Should explain notions of paramagnetic and ferromagnetic phases, what exactly the SK model is, how large sigma corresponds to low temperature, what is meant by "transitory dynamics" and how this relates to criticality, what it means to not find a phase transition between the paramagnetic and ferromagnetic phases, and the discussion text on chaos in different regimes.

To address the reviewer's concerns, we modified the text as follows:

- First paragraph in the Results section: We removed the mentioning to the Hamiltonian, as it is not necessary and would confuse biology readers who may not know what the Hamiltonian is.
- First paragraph in the Results section: We replaced "To fit the model," by "Because the model assumes binary data,".
- We added the following text in the sentence right after equation (2).
"Although we refer to E as the energy, E does not represent the physical energy of a neural system but is a mathematical construct representing the frequency with which activity pattern \mathbf{S} appears in the given data. Activity pattern \mathbf{S} appears rarely in the data if E corresponding to \mathbf{S} is large and vice versa. Parameter h_i represents the tendency that $S_i = 1$ is taken because a positive large value of h_i implies that $S_i = 1$ as opposed to $S_i = -1$ lowers the energy and hence raises the probability that \mathbf{S} with $S_i = 1$ appears. Parameter J_{ij} represents a functional connectivity between ROIs i and j because, if J_{ij} is away from 0, S_i and S_j would be correlated in general."
- Right after Eq. (3): We supplied the definition of the SK model.
- Right after Eq. (3): We removed a sentence mentioning the relationship between σ and the temperature, because it is unnecessary to understand this connection and because it was slightly inaccurate in its original form due to the μ term in Eq. (3).
- First half of the third paragraph in the Results section: We added text to enhance the explanation of m , q , and each of the three phases.
- Third paragraph in Introduction: To explain what is meant by transitory dynamics, we added "(i.e., dynamic transitions among discrete brain states)".

To better explain "what it means to not find a phase transition between the paramagnetic and ferromagnetic phases" to non-physicists, we added the following text in the Discussion section (p. 6, lines 264-265).

“Roughly speaking, paramagnetic and ferromagnetic phases correspond to active and quiescent phases, respectively.”

This added text together with the surrounded text explains that the transitions we found are not the ones between quiescent and active phases. We decided not to add similar explanation in the previous paragraph in the Discussion section (i.e. paragraph starting with “There are various types...”), which the reviewer probably pointed to. This is because it is difficult to explain what paramagnetic and ferromagnetic phases intuitively mean without referring to neural avalanches (as we did in the next paragraph). We considered that adding such discussion in the mentioned paragraph would rather confuse non-physicist readers than to aid their understanding.

We attempted to improve “the discussion text on chaos in different regimes” in the latter half of the paragraph starting with “There are various types...” in the Discussion section. However, we opt not to extend this part because this is a specialist discussion anyways, if we get into details. As written in the present text, there are different types of chaos, which is difficult to be explained in intuitive terms.

4. p4, 2nd para, bit incongruous to refer to $r=0.21$ as “mildly correlated” versus $r=0.24$ on the previous page being referred to as “significantly positively correlated”.

We replaced “significant” by “mild” (p.4, line 142).

5. p5, the discussion of how these results are consistent with balanced excitation and inhibition is unclear and unconvincing.

We removed the entire paragraph.

6. p5, re discrepancy with studies finding ferromagnetism in fMRI data, the explanation given in terms of the present research using fMRI data seems like faulty logic (same type of data no?) How do the results reconcile with those of e.g. Ref. 55?

We resolved the inconsistency with the results in Fraiman et al. (2009) (Ref. [62] in the revised manuscript) by rewriting the text as follows:

“Fraiman *et al.* reported that the Ising model at the paramagnetic-ferromagnetic phase transition explains properties of the functional networks based on fMRI data [Fraiman2009]. They used a two-dimensional Ising model with a uniform strength of interaction between pairs of nodes that are adjacent on a square lattice (and $J_{ij} = 0$ for the rest of pairs). Another study that suggested the paramagnetic-ferromagnetic phase transition for fMRI signals also assumed a uniform J_{ij} [Kitzbichler2009]. In contrast, we did not constrain the J_{ij} values and instead inferred the J_{ij} values (i.e., structure of functional network) using the PMEM. Because these previous studies [Fraiman2009, Kitzbichler2009] did not assume heterogeneity in J_{ij} as we did, their results do not contradict ours. In fact, the assumption of a uniform J_{ij} corresponds to setting $\sigma = 0$ in our phase diagrams. If one varies μ under the condition $\sigma = 0$, the only possible phase transition is the paramagnetic-ferromagnetic transition (Fig. 1a–d). However, that phase transition point under the condition $\sigma = 0$ is far from the location of the empirical data when σ is allowed to deviate from 0 (crosses in Fig. 1a–d). Therefore, allowing heterogeneity in J_{ij} may be key to further clarifying the nature of critical neural.”

As a side note, we found that it was inappropriate to refer to Marinazzo et al. (2014) here for supporting the ferromagnetism in fMRI data because they used a structural network, not a functional network, on which to run an Ising model. Therefore, we changed “Computational studies also support the ferromagnetism of fMRI data [Fraiman2009, Kitzbichler2009, Marinazzo2014]” to “Computational studies also support the ferromagnetism [Fraiman2009, Kitzbichler2009, Marinazzo2014].”

Responses to Reviewer #4

We appreciate the reviewer's useful suggestions. We have addressed all the comments by the reviewer. The page numbers in our responses refer to those in the revised manuscript unless otherwise mentioned. The deleted and added text is shown in **red** and **blue**, respectively, in the revised manuscript.

In the present article, Ezaki et al. used a maximum entropy model to map the whole-brain BOLD dynamics onto a phase diagram. They showed that dynamics were poised in the paramagnetic phase and close to a spin-glass phase transition. They then showed that subjects with larger performance IQ presented dynamics that were closer to that phase transition.

I enjoyed reading the article, which is clearly written. The method presented therein will be very useful for the neuroscience community and, thus, I think that the code to learn the model should be made publicly available. However, I have some concerns that should be addressed in order to improve the article:

We are glad to hear the overall positive evaluation by the reviewer.

Previously we published the code for learning the PMEM [Ezaki2017] (Ref. [38] in the revised manuscript). We provided this information in the revised manuscript as follows (p. 8, lines 338-339):

“We previously provided MATLAB code for estimating the Ising model from data by PL maximization [Ezaki2017]. The code is publicly available on GitHub repository (<https://github.com/tkEzaki/energy-landscape-analysis>).”

1) My first concern relates to the binarization of the fMRI data. The authors binarized the data by first using z-score and setting a value equal to 1 if fluctuations were positive and -1 otherwise. Thus, by construction of the binary data, the magnetization must be $m = 0$, forcing the system to be in the paramagnetic phase. I think that taking the standard deviation as the threshold (as it is the common practice) would imply $m > 0$. In figure S2, the authors varied the binarization threshold, but they only showed the

correlation between the IQ and spin-glass susceptibility (which depends on the covariance of the data). The authors should show how the point representing the data in the phase diagram changes with different thresholds.

We computed the phase diagrams with two different binarization threshold values, $\theta = 1$ and $\theta = -1$. Note that, with these threshold values, the fraction of $S_i = +1$ is equal to ~ 0.148 and ~ 0.853 , respectively (see the caption of Fig. S2). Please note that, although the m_i values and hence the estimated h_i values are not close to zero with the new thresholds, we forced $h_i = 0$ for the purpose of calculating the phase diagram. This is because the phase diagram for physical magnetic systems is concerned with what happens when the external magnetic field (i.e., h_i) is turned off [Fisher1991, Reichl1998]. The results are shown in new Supplementary Fig. 3. We found that the phase diagrams and the position of the point representing the data in the phase diagrams are qualitatively the same as those reported in the main text except a notable difference, which is not detrimental to our main claim. The major difference is that the point representing the data is roughly on the boundary between the paramagnetic and the SG phases with the new thresholds. This result is in fact consistent with the brain criticality hypothesis.

We mentioned these robustness results in the main text (p. 4, lines 156-157) and explained the detail in Supplementary Fig. 3.

[Reichl1998] Reichl, L. E. *A Modern Course in Statistical Physics*. (Wiley, New York, 1998).

[Fisher1991] Fischer, K. H., Hertz, J. A. *Spin Glasses*. (Cambridge University Press, Cambridge, 1991).

2) Also, it would be interesting to compare the results with those from surrogate data. Specifically, the authors could generate multivariate gaussian time-series with the same covariance as the empirical data, apply the threshold, and see where the surrogate data are placed on the phase diagram. My guess is that the surrogates would be placed close to the spin-glass transition. Moreover, the data from subjects with large and low IQ could be classified using a simple Gaussian decoder, based on the covariance of the data. If this simple decoder achieves better discrimination than using the model parameter sigma, the results would be weakened. In other words, if a simple model such as a multivariate Gaussian process leads to better discrimination and relatively

same working point in the phase diagram, then the proposed model is somehow redundant (and conceptually more complex).

Technically speaking, we agree with all the reviewer says here. However, the aim of this paper is not to develop an efficient decoder but to provide empirical support of the critical brain hypothesis. We added the following new paragraph to the Discussion section to discuss this issue and articulate the aim of the present study.

“One could classify the data from participants with high and low IQ scores using a simple multivariate Gaussian decoder [Bishop2006]. Such a decoder would assume as input the mean and covariance of the fMRI data for each participant or its random samples having the same mean and covariance. Because our PMEM also assumed the same input but was not optimized for classifying the participants, an optimized Gaussian decoder will probably be more efficient than our PMEM in explaining the IQ scores of the participants. This approach is conceptually much simpler than the present one, which employ the PMEM and its phase diagrams. However, the aim of the present study was to find empirical support of the critical brain hypothesis by relating the fMRI data to the phase diagrams of a prototypical spin system rather than to efficiently classify participant.”

3) The maximum entropy model was built using the binary data of all subjects. Because the data could be highly variable between subjects, the model could be biased to replicate this diversity. To test this, the authors should learn two models using the halves of the data, test the model built from the first-half data on the second-half data and vice versa, and evaluate the likelihood ratios (train vs test). Likelihood ratios close to 1 would indicate that the model can generalize.

We addressed this issue in the revised manuscript but did not use the likelihood ratio for the following reasons. The pairwise maximum entropy model (PMEM) adjusts the $\langle S_i \rangle$ and $\langle S_i S_j \rangle$ values to those of the empirical data and leaves higher order correlations unassumed. Although the two halves of the data had similar pairwise correlation values (Fig. S5), the pairwise correlation values were not exactly the same between the halves, reflecting the heterogeneity in participants. This extent of similarity/dissimilarity in the correlation structure is inherited to the estimated PMEMs.

The likelihood ratio suggested by the reviewer inevitably deviates from 1.0 and practically does so to a large extent even if the data sets are generated from exactly the same model [Hoel1984].

In fact, the likelihood ratios computed as follows were not standardized values comparable to unity:

$$\frac{\mathcal{L}_{\text{test}}^1}{\mathcal{L}_{\text{train}}^1} = \exp\left(\sum_{S^1} E^1(S^1) - \sum_{S^2} E^1(S^2)\right) = \exp(-163960.969),$$

$$\frac{\mathcal{L}_{\text{test}}^2}{\mathcal{L}_{\text{train}}^2} = \exp\left(\sum_{S^2} E^2(S^2) - \sum_{S^1} E^2(S^1)\right) = \exp(-134162.283),$$

where $\mathcal{L}_{\text{test}}^i$ and $\mathcal{L}_{\text{train}}^i$ ($i = 1, 2$) are the likelihood functions of the models estimated for the i -th half of the data, which are calculated for the test data and train data, respectively. However, this does not mean that the model cannot generalize [Hoel1984]. To directly assess the generalizability of the model, we carried out the following cross validations.

First, as the reviewer suggested, we split the participants into two subgroups and estimated PMEMs for each subgroup. We confirmed that the models predicted the correlation structure in the other subgroup with a reasonable accuracy (Fig. S5). Second, the phase diagrams estimated separately for the two subgroups were similar to each other and to the phase diagram estimated for the set of all the participants (Fig. S6). Finally, we estimated $\tilde{\mu}$ and $\tilde{\sigma}$ (as we did in Fig. 2a) for each participant using the phase diagrams estimated for the subgroup of half the participants that the focal participant belonged to. The results were highly consistent with those reported in the main text produced using all the participants (Fig. S7). Collectively, we concluded that the estimation error caused by a finite number of participants did not considerably affect our main results.

We added a subsection (with heading “Effects of data length and individual variability”) in the Results section to explain these results.

[Hoel1984] Hoel, P.G. *Introduction to mathematical statistics*. (Wiley, New York, 1984)

- 4) The mapping of the data in the phase diagram shows rather that the system is subcritical. Recent studies showed that subcritical dynamics are more likely during rest (Priesmann et al., 2014; Hahn et al., 2017), with potential functional advantages. This

should be discussed. Note also that, in the finite size scaling analysis, the susceptibility grows with N' but there is no sign of divergence.

Thanks for drawing our attention to these important references. We added the following text to discuss these references in the Discussion section.

“We showed that neural dynamics for each participant were close to but substantially off the criticality separating the paramagnetic and SG phases. Other studies using the PMEM [Hahn2017] and other models [Priesemann2014] also support off-critical as opposed to critical neural dynamics in the brain. The study applying the PMEM to local field potentials suggested that such off-critical dynamics may potentially have functional advantages because the off-critical situation would prevent the dynamics to get past the phase boundary to enter the other phase under the presence of noise [Priesemann2014]. The other phase may correspond to pathological neural dynamics such as epilepsy. The off-critical neural dynamics that we found for our participants, regardless of their IQ scores, may benefit from the same functional advantage.”

5) I am not familiar with IQ studies, but I think that sentences such as “more intelligent human individuals” or “more intelligent human brains” are not appropriate and should be avoid.

Thanks for pointing this out. We changed the text as follows:

- Abstract, line 6: “more intelligent human participants” -> “human participants with higher intelligence quotient scores”
- Introduction, last paragraph, line 1: “different degrees of intelligence” -> “different intelligence quotient (IQ) scores”
- Introduction, last line: “more intelligent human individuals in the sense of” -> “individuals higher in the intelligence score that measures”
- Discussion, first line: “more intelligent human brains” -> “neural dynamics of humans with higher intellectual ability”
- Discussion, fourth paragraph, second line: “intelligent performance” -> “intellectual ability”
- Discussion, fourth paragraph, fourth line: “the intelligence” -> “the intellectual score”

References:

V. Priesemann, et al., "Spike avalanches in vivo suggest a driven, slightly subcritical brain state," *Front. Syst. Neurosci.*, 2014.

Hahn G et al. (2017) Spontaneous cortical activity is transiently poised close to criticality. *PLoS Comput Biol* 13(5): e1005543.

I hope the authors find this comments useful.

REVIEWERS' COMMENTS:

Reviewer #3 (Remarks to the Author):

The authors have done an excellent job in revising the manuscript, and have responded thoughtfully to all concerns.

Reviewer #4 (Remarks to the Author):

The authors have correctly addressed all of my concerns, except my 2nd question. I would like to see where Gaussian surrogates are placed in the phase diagram. Apart for this important point, the authors did a good job responding my comments and the paper was highly improved.

Responses to Reviewer #4

We appreciate the reviewer's useful suggestions. We have addressed all the comments by the reviewer. The page numbers in our responses refer to those in the revised manuscript unless otherwise mentioned. The deleted and added text is shown in **red** and **blue**, respectively, in the revised manuscript.

The authors have correctly addressed all of my concerns, except my 2nd question. I would like to see where Gaussian surrogates are placed in the phase diagram. Apart for this important point, the authors did a good job responding my comments and the paper was highly improved.

We are glad to hear the overall positive evaluation by the reviewer.

We computed χ_{SG} and χ_{uni} for the Gaussian surrogate for each participant and mapped them on the phase diagram as we did in Fig. 2a in the main text. As expected, the results were similar to those for the original empirical data. We included these results in new Supplementary Figure 9 and referred to them in the discussion section in the main text.

Supplementary Figure 9. Results obtained with surrogate data. **a** Comparison between χ_{SG} computed for the empirical data and that for the surrogate data. **b** Comparison between χ_{uni} computed for the empirical data and that for the surrogate data. **c** Distribution of the participants on the phase diagram computed for surrogate data (see also Fig. 2a in the main text). To obtain the surrogate data for each participant, we observed $t_{max} = 2 \times 10^4$ samples from a multivariate Gaussian distribution having the same mean vector and covariance matrix as those for the empirical data. Each circle represents a participant. These results suggest that our main results are reproduced solely from the covariance structure in the empirical data.